# The Role of Serine-Threonine Protein Phosphatase PP2A in Plant Oxidative Stress Signaling—Facts and Hypotheses

**DOI:** 10.3390/ijms20123028

**Published:** 2019-06-21

**Authors:** Csaba Máthé, Tamás Garda, Csongor Freytag, Márta M-Hamvas

**Affiliations:** Department of Botany, Faculty of Science and Technology, University of Debrecen, Egyetem tér 1., H-4032 Debrecen, Hungary; gtamas0516@gmail.com (T.G.); fcsongor@me.com (C.F.); hamvas.marta@science.unideb.hu (M.M-H.)

**Keywords:** reactive oxygen species, plant oxidative stress, protein phosphatase, PP2A, ROS signaling pathways

## Abstract

Abiotic and biotic factors induce oxidative stress involving the production and scavenging of reactive oxygen species (ROS). This review is a survey of well-known and possible roles of serine-threonine protein phosphatases in plant oxidative stress signaling, with special emphasis on PP2A. ROS mediated signaling involves three interrelated pathways: (i) perception of extracellular ROS triggers signal transduction pathways, leading to DNA damage and/or the production of antioxidants; (ii) external signals induce intracellular ROS generation that triggers the relevant signaling pathways and (iii) external signals mediate protein phosphorylation dependent signaling pathway(s), leading to the expression of ROS producing enzymes like NADPH oxidases. All pathways involve inactivation of serine-threonine protein phosphatases. The metal dependent phosphatase PP2C has a negative regulatory function during ABA mediated ROS signaling. PP2A is the most abundant protein phosphatase in eukaryotic cells. Inhibitors of PP2A exert a ROS inducing activity as well and we suggest that there is a direct relationship between these two effects of drugs. We present current findings and hypotheses regarding PP2A-ROS signaling connections related to all three ROS signaling pathways and anticipate future research directions for this field. These mechanisms have implications in the understanding of stress tolerance of vascular plants, having applications regarding crop improvement.

## 1. Introduction

Reactive oxygen species (ROS) are produced in all eukaryotic cells as a consequence of both internal metabolic events and external stressors. For plants, this includes a wide range of abiotic factors (drought, extreme temperatures, high salinity, mechanical stress, UV exposure, etc.) and biotic factors like pathogens, where their diverse metabolites are the triggering factors [1]. The main ROS produced as a consequence of metabolic processes are singlet oxygen (^1^O_2_), superoxide anion (O_2_^•−^), and hydrogen peroxide (H_2_O_2_). In addition, atmospheric ozone (O_3_) can be transformed intracellularly or in the apoplastic space into hydroxyl radical (OH^•^) or perhydroxyl radical (O_2_H^•^) [2]. For the hydroxyl radical, another major site of production is the PSI of chloroplast thylakoids via the Fenton reaction [3]. For many cellular events, ROS are thought to be natural consequences of metabolism or even beneficial as signaling molecules e.g., for mitosis [4]. However in general, they are known to induce cell damage by altering the redox status of cells, oxidating proteins and lipids, and inducing DNA breakage [2,4]. Meanwhile, the so-called “oxidative burst” characterized by local oxidative damage following pathogen attack, leads to cell death in the affected tissues that isolates the pathogen from the rest of/healthy tissues [5].

In order to avoid negative effects of ROS, plants developed multiple scavenging systems. This involves non-enzymatic compounds- α-tocopherol, ascorbic acid, polyphenolics, carotenoids, flavonoids, reduced glutathione- and enzymatic scavengers: e.g., catalases (CATs), superoxide dismutases (SODs), peroxidases (PODs), peroxiredoxins (PrxR) and glutathione reductases (GRs), part of them being involved in the so- called ascorbate-glutathione and glutathione peroxidase cycles. Ascorbate peroxidase (APX) is a plant-specific enzyme (for comprehensive description [1,2]).

The main ROS producing systems in plants are photosynthetic and respiratory electron transport, plasma membrane NADPH oxidases, and glycolate oxidase functioning during photorespiration [6]. The cellular/subcellular localizations of ROS producing and scavenging are as follows [2,6]:(i)Plasma membrane-associated NADPH oxidases (respiratory burst oxidase homologs, RBOHs). These will produce superoxide radicals during the formation of NADP^+^, which are then reduced to H_2_O_2_ by SOD. The nature and localization of this SOD isoform was controversial for a long time [7]. More recent research shows it is Cu/Zn SOD [8].(ii)For chloroplasts, the thylakoid membrane associated photosynthetic electron transport chain produces singlet oxygen at PSII and superoxide both at PSII and PSI (the latter by the Mehler reaction). Superoxide anion is then scavenged to H_2_O_2_ by Cu/ZnSOD and FeSOD. Hydrogen peroxide is then reduced by APX and the ascorbate-glutathione cycle.(iii)In mitochondria, superoxide is produced at complexes I (NADH dehydrogenase) and III (ubisemiquinone) of the inner membrane and scavenged to H_2_O_2_ by MnSOD. Components of the ascorbate-glutathione cycle from both the intermembrane space and matrix scavenge hydrogen-peroxide.(iv)In peroxisomes, H_2_O_2_ is produced via the activity of glycolate oxidase during photorespiration and the β-oxidation of fatty acids. O_2_^•−^, also produced here is scavenged by Cu/ZnSOD and MnSOD, while H_2_O_2_ is scavenged by catalases and APX [9,10].(v)In the cytosol, superoxide anions produced by diverse metabolic processes are scavenged by Cu/ZnSOD and then by the ascorbate-glutathione cycle.(vi)Cell wall/ apoplastic space. ROS producing and scavenging directly at the level of cell wall is not well known, but in general terms of apoplast, ROS produced mainly by the activity of NADPH oxidases will lead ultimately to a systemic oxidative burst by their cell-to-cell spread as seen during pathogen attack. During ozone exposure, the ROS produced will be used for reducing ascorbate to dehydroascorbate (DHO) that regulates intracellular ROS signaling [11]. Moreover, class III peroxidases that are anchored to the cell wall, may also contribute to apoplastic H_2_O_2_ generation [12].

ROS induce cellular/subcellular changes directly by inducing damage of macromolecules essential for life, or- since they may be important signaling molecules- by diverse signal transduction pathways. The aim of the present review is to give an insight into some ROS signaling mechanisms. According to present knowledge there are three main pathways of ROS signaling, with many known and possible interconnections between them. Additionally, ROS triggers signaling pathways that will increase the level of non-enzymatic antioxidants and increases the expression of genes coding enzymatic antioxidants:(i)Pathway 1. External ROS, mainly superoxide and H_2_O_2_ produced by RBHOs trigger signal transduction pathways that alter gene expression leading to oxidative stress responses (see references [2,13,14] for an examples). External ROS may arise by the “ROS wave mechanism”: the activity of RBOH results in the production of extracellular ROS or pathogens induce the formation of intracellular ROS that is released and stimulates ROS formation in the neighboring cells [15].(ii)Pathway 2. Intracellular ROS generated by distinct cell compartments (see above) is extensively studied and this can trigger signaling pathways as well.(iii)Pathway 3. From the perspective of the present review, this is probably the most exciting pathway. Here, environmental stresses induce signal transduction pathways that lead to ROS generation by induction of the expression of ROS generating enzymes, then ROS generated by this mechanism induces further intracellular changes involving Pathways 1 and 2 (see reference [16] for an example). ABA is a well-known mediator of intracellular H_2_O_2_ generation in guard cells [13] and Pathway 3 has a crucial role in this (Subchapter 3.3.).

All these pathways involve protein phosphorylation dependent steps, mainly the mitogen activated protein kinase (MAPK) cascades. Since many of these kinases phosphorylate proteins at serine/threonine residues in eukaryotes including plants [17,18], it is obvious that the counteracting serine-threonine protein phosphatases are involved in these pathways. The metal- dependent phosphatases (PP2C) are well-known to be involved in ROS-related signaling e.g., during an ABA-mediated oxidative response [13]. Meanwhile, the metal-independent protein phosphatase family, PP2A is the most abundant and perhaps the most important Ser/Thr protein phosphatase in eukaryotic cells [19]. However, there is still relatively little knowledge on its involvement in ROS-related signaling. The aim of the present review is to present current knowledge and working hypotheses on its role in ROS signaling and scavenging. The recently published, excellent review of Bheri and Pandey [20] gives several examples of PP2A-oxidative stress relationships. The next section will provide a survey on the basic types and functions of serine-threonine protein phosphatases, as well as the ROS-related developmental, cellular and molecular effects of PP2A inhibitors in vascular plants. Finally, the main section (Section three) will describe the involvement of PP2A in the signaling pathways listed above.

## 2. Types of Serine-Threonine Protein Phosphatases and ROS-Related Effects of PP2A Inhibitors in Higher Plants

### 2.1. Overview of Serine-Threonine Phosphatases with Special Emphasis on PP2A

These phosphatases (PSP) catalyze the dephosphorylation of phosphoserine/phosphothreonine side chains of proteins. They represent a wide family of enzymes having important roles in practically every event of eukaryotic cell life. This includes signal transduction pathways, e.g., those involving MAPK cascades, the regulation of cell cycle and metabolism, stress responses and defense, etc. [19,21]. PSPs include three major families: phosphoprotein phosphatases (PPPs), metal-dependent phosphatases (PPMs) and aspartate-dependent phosphatases. PPPs are then subdivided to PP1, PP2A, PP2B (calcineurin, dependent on Ca^2+^, calmodulin and Mg^2+^), PP4, PP5, PP6 and PP7 [19]. The main member of PPM is the Mn^2+^/Mg^2+^ dependent PP2C [19]. PP2C is the main phosphatase known for its involvement in plant ROS signaling [13,22], but the present review does not concentrate primarily on this enzyme. Even though it is not subject of the present study, it is worth mentioning that dual specificity (both Ser/Thr and Tyr) MAPK phosphatases have been reported to be involved in plant stress signaling [23].

Regarding enzymes of the PPP family, to date, there is less evidence for the involvement of PP1 in plant oxidative stress events. Regarding the Ca^2+^-calmodulin dependent PP2B, its presence was not proven to occur in plants until recently [24]. PP4-7, of which PP4 and PP6 are closely related to PP2A, are of relatively minor occurrence and importance in plant stress signaling. Thus, we give a more detailed description only for PP2A, probably the most important PPP in eukaryotes and particularly, in plants.

In *Arabidopsis*, there are five catalytic subunit isoforms of PP2A, much less, than the number of serine-threonine protein kinase catalytic subunits [24,25]. Meanwhile, PP2A has multiple functions in plant cells: growth and stress related signaling including hormone-related signal transduction pathways, cell cycle regulation, vesicle trafficking, auxin transport, as well as regulation of the activities of a huge number of enzymes involved in key metabolic pathways [25]. Moreover, these events are of diverse subcellular localizations. How is this possible with such a low number of catalytic subunit isoforms? PP2A is in fact a heterotrimer, with the core enzyme consisting of a scaffolding, A subunit PP2A/A (3 isoforms of around 65 kDa in *Arabidopsis*) and the catalytic, C subunit PP2A/C (PP2Ac, isoforms are of around 36 kDa)- and the variable B subunit PP2A/B with at least 17 isoforms [24,25]. As in other eukaryotes, these plant B subunits are further subdivided into B, B’ and B” with a wide range of molecular weights (54–130 kDa) [19,25]. One of the most well-known proteins is TON2/FASS of *Arabidopsis*, that is a B” subunit with crucial roles in microtubule organization both in mitotic and non-mitotic plant cells [26,27]. The B subunits are responsible for the multiple localizations and functions of PP2A complexes. Moreover, if a single protein contains multiple phosphoserine/threonine dephosphorylation sites, they can modulate the overall catalytic activity of the complex and determine, which sites will be modified and which amino acids will remain unchanged [28]. B regulatory subunits are very important in mediating the effects of PP2A inhibitors, as we will see in the next section. They are strongly related to the regulation of oxidative stress responses as well. For example, an *Arabidopsis* PP2A B’ subunit allows activity of PP2A at the proper subcellular sites only when the pathogen is absent. During a pathogen attack, this subunit may be inactivated, which decreases PP2A activity and leads to the pathogenesis response involving H_2_O_2_ production (see subchapter 3.3.) [29].

Table 1. shows some characteristic examples of PP2A subunits with potential or actual functions in oxidative stress response/signaling in plants. The relevant functions of these subunits will be analyzed in detail in Chapter 3.

### 2.2. How PP2A Inhibitors Influence ROS Production and Signaling in Plants?

There are several inhibitors of PP2A that potently and specifically inhibit the activity of this enzyme family [41,42] These inhibitors are tools that are frequently used for the study of the structure and functioning of PP1 and PP2A [21,43]. Practically all of the most frequently used inhibitors affect both protein phosphatase activities and ROS production in plants. These are: microcystin-LR (MCY-LR), calyculin A (CA), okadaic acid (OA) and cantharidin. Another potent PP2A inhibitor is endothall [41,44,45], but its phosphatase related effects on oxidative stress are little known for plants. The effects of inhibitors will be described in the following sections. The known and potential oxidative stress related targets of these inhibitors are shown on Figure 1.

#### 2.2.1. Microcystin-LR

MCY-LR is one of the most extensively studied and used phosphatase inhibitors, mainly for two reasons: (i) it is produced by many cyanobacterial strains during eutrophication of fresh- and even marine waters. Because of its toxicity for aquatic plants and animals, it may cause serious damage in aquatic ecosystems and may influence human health (see reference [46] for an example). (ii) it is widely used as a tool for the study of protein phosphatases (see reference [43] for an example). It is a cyclic heptapeptide, a potent inhibitor of both PP1 and PP2A. MCY-LR binds covalently to the active site of catalytic subunits, making the substrates—phosphoserine- and –threonine side chains- unavailable to the enzyme. Thus, in contrast to drugs like OA, MCY-LR mediated phosphatase inhibition is irreversible [41,42]. For in vitro assays, MCY-LR inhibits these phosphatases with similar potency, with 50% inhibitory concentrations (IC_50_) in the subnanomolar range [47]. In vivo, the degree of PP2A inhibition depends on the species and the tissue studied. For example, a relatively low concentration (0.2 µM) of MCY-LR inhibits PP1 more potently, than PP2A (there is even a slight stimulation of PP2A activity) in tips of lateral roots, while it slightly inhibits both phosphatases in extracts of whole lateral roots of *Vicia faba* (broad bean) [48]. Higher MCY-LR concentrations severely inhibit the activities of both PP1 and PP2A, irrespective of the studied tissue. These changes were attributed to differential effects of the inhibitor on the B regulatory subunits [48]. Non-canonical regulatory subunits may play important regulatory functions for PP2A as well. For example, α4 forms complexes with PP2A/C, independently to A and B subunits [49]. Low MCY-LR concentrations increase PP2A activity in the human embryonic kidney cell line HEK293, but inhibit this activity in the human liver cancer cell line SMMC-7721. These differences were induced by different binding ability of α4 to the catalytic subunit of PP2A, in the presence of MCY-LR. Moreover, the activity of the PP2A/A-PP2A/C complex depends on its own phosphorylation state [50]. These in vivo effects of MCY-LR may be important for understanding of its oxidative stress inducing activity as well.

Many effects related to oxidative stress responses were reported for MCY-LR in vascular plants. Most of the studies reveal that it increases ROS levels (e.g., in rice roots), reduces the levels of reduced glutathione and increases the activities of antioxidant enzymes like SODs, PODs, CATs in aquatic plants as well as in model plants like white mustard (*Sinapis alba*), broad bean (*Vicia faba*) and *Arabidopsis* [51,52,53,54,55].

There are several models for the mechanisms of ROS induction by MCY-LR in eukaryotes (Figure 1).

(i)Mechanism 1. MCY-LR generates ROS directly, without the involvement of protein phosphatases. The most probable pathway is the formation of MCY-LR-glutathione conjugates catalyzed by glutathione-*S*-transferase (GST) leading to the decrease of reduced glutathione (GSH) levels. This leads to oxidative stress –ROS production- by inducing an imbalance in the cell redox state, leading to lipid peroxidation, cytoskeletal disruption, mitochondrial damage, nuclear DNA strand breaks. Oxidative stress modulates expression and activities of antioxidant enzymes as well [56,57]. Based on assays of MCY-glutathione conjugate levels/ GST activities and of antioxidant enzyme activities, this model was proposed in plants by Pflugmacher [58,59]. Related to this, the level of lipid peroxidation as proven by the elevation of malondialdehyde level, increased in MCY-LR treated pepper fruits and *Arabidopsis* seedlings [53,60]. It is worth mentioning that MCY-LR can bind purified catalase that changes its activity [61], suggesting another direct effect of the inhibitor on oxidative stress.(ii)Mechanism 2. MCY-LR induces signaling pathways leading to ROS production without evidence for the involvement of the protein phosphatase PP2A. Related to this, the inhibitor modulated Ca^2+^ dependent signaling pathways involving calcium calmodulin dependent multifunctional protein kinase II (CaMKII) to produce ROS [62]. Similar pathway has not been proven for plants.(iii)Mechanism 3. MCY-LR induces signaling pathways, leading to ROS production with proven or possible involvement of PP2A. MCY-LR induces cytoskeletal disruption via a MAPK28 pathway that can be modulated both by protein phosphorylation and oxidative stress in the nervous system, however the mechanisms of the connection between these two cellular processes were not clearly elucidated [63]. For *Brassica rapa*, MCY-LR induced ROS formation by NO formation which is known to release Ca^2+^ at the intracellular level. Since NADPH oxidase expression is protein phosphorylation dependent, this probably initiates a phosphorylation dependent cascade, leading to the regulation of NADPH oxidase which is responsible for ROS production [64]. Gehringer [65] raised the possibility that the inhibition of PP2A by MCY-LR activates MAPKs and among other subcellular changes, this might lead to the production of ROS. MCY-LR prevents dephosphorylation of the multifunctional Ca^2+^/calmodulin-dependent protein kinase (CaMKII) at Thr286, leading to its activation and finally, cell death of rat hepatocytes [66]. These data and hypotheses raise the possibility of the usefulness of MCY-LR as a tool for further study of phosphatase mediated ROS production in plants.

Overall, although many studies report MCY-LR mediated oxidative stress in plants, only a few publications attempted to explain the biochemical mechanisms laying behind (see Figure 1 for a summary). Several studies show that uptaken MCY-LR will be conjugated to glutathione as a detoxifying mechanism. There is no direct evidence for the MCY-LR mediated ROS induction via protein phosphatase dependent pathways. However, there is evidence that a significant fraction of MCY-LR is not conjugated intracellularly: it significantly inhibits the activities of PP1 and PP2A in *Sinapis alba*, *Phragmites australis* and *Vicia faba* [52,67,68]. The complex effects of the inhibitor on both catalytic and regulatory subunits of PP2A are very likely to interfere with oxidative stress mediated signaling pathways. Further research is needed to clarify this issue.

#### 2.2.2. Calyculin A and Cantharidin

CA is a phospholipid polyketide from the marine sponge *Discodermia calyx* [41] As MCY-LR, CA inhibits PP1 and PP2A with similar potency in vitro [42]. For the human breast cancer cell line MCF7, this is the effect in vivo as well [69]. However, for mouse ventricular myocytes, it inhibits PP2A more potently than PP1 [70], indicating that the in vivo effects of CA are cell type dependent in mammals, due to the differences in regulatory subunits of protein phosphatases. For plants, we did not find significant data for such differences. Chandra and Low [71] have shown that inhibition of protein phosphatase activities by CA and OA induced oxidative burst mimicking the effects of pathogens in soybean. Since they did not measure PP1 and PP2A activities, we do not know which type of phosphatase was primarily affected. The authors did not look for NADPH oxidase activities, but it is known this ROS producer is one of the main inducers of oxidative burst [2]. However, the study of Chandra and Low [71] has already suggested that oxidative stress induction occurs, when proteins involved in stress signaling remain in a phosphorylated state. Treatment of tobacco suspension cells with CA activated the 50 and 46 kDa MAP kinases, leading to oxidative burst. These MAPKs are tyrosine kinases, but they are activated when certain Ser/Thr residues remain in the phosphorylated state [72]. What type of protein phosphatase plays a primary role in this? Cantharidin, that inhibits PP2A more potently, than PP1, has a similar effect to CA [72]. Thus, PP2A is likely to be involved in the above process. An interesting study showed that CA mimicked the stress-promoting effects of fungal elicitors (known to induce oxidative stress) via protein phosphatase inhibition and the consequent maintenance of proteins in a phosphorylated state [73]. One of the most interesting studies related to the effects of CA in plants showed that this inhibitor overstimulates the phosphorylation of OsPHB protein in *cdr1* lesion mimic rice mutants, and this is leading to cell death [74]. OsPHB is a homologue of prohibitin, a senescence/cell death related protein in yeast and mammals. This is a chaperone playing a role in the assembly of proteins of mitochondrial respiratory chain complexes. CDR proteins are decreasing the phosphorylation state of NADPH oxidase and maintaining the dephosphorylated state of PHB. This form of the PHB protein protects mitochondrial membranes, mainly the proteins of inner membrane from lesion. The effects of CA- further increasing the phosphorylation state of PHB in the *cdr1* mutant- show that protein phosphatases play a crucial role in this and that in the absence of normal PP2A (and/or) PP1 activity, ROS production will be induced either by the activation of NADPH oxidase, or by inducing mitochondrial permeability transition (MPT). The involvement of proteolysis cannot be excluded in this cell death pathway [74]. Moreover, the regulation of NADPH oxidase by phosphorylation/dephosphorylation and its involvement in protein phosphatase inhibitor induced ROS production has been proven previously by this research team: the NADPH oxidase inhibitor DPI decreased the CA induced high H_2_O_2_ levels in cell suspension cultures of the *cdr1* and *cdr2* mutants of rice. Cantharidin and OA, inhibitors that affect PP2A more potently than PP1, did not induce ROS production, suggesting that PP1 was the phosphatase that is primarily involved in this process [75]. However, other studies have shown the crucial role of PP2A, too in this process (see Chapter 3 as well). The respiratory burst NADPH oxidases are subject of direct phosphorylation/dephosphorylation as shown by increase of the phosphorylation state of *Arabidopsis* rbohD (AtrbohD) transfected to HEK293T cells, in the presence of CA *in vivo*. This study showed that both Ca^2+^ binding and phosphorylation activates ROS production by AtrbohD in a synergistic way [76]. In this study, CA could maintain phosphorylation of RBOH either by acting on a phosphatase that directly dephosphorylates it, or through indirect mechanisms like modulation of phosphorylation dependent signaling pathways.

All the research presented above shows that CA is a valuable tool for studying the connection between PP2A (and PP1) and oxidative stress.

Cantharidin is a terpenoid produced by blister beetles [42,77]. As we mentioned previously, it inhibits PP2A more potently than PP1 both in vitro and in vivo [42]. An interesting study shows that despite the phosphatase activity inhibitory effects, Western blot demonstrated that cantharidin increased the expression of PP2Ac after short-term (2 h and 10 h) exposure and decreased its level only after 24 h of exposure of *Arabidopsis* seedlings [78]. Cantharidin treatment increases the levels of many proteins, most of them being localized in chloroplasts. One such protein family is the chloroplast and non-chloroplast GSTs, important detoxification enzymes [78]. Whether cantharidin, as MCY-LR is directly detoxified by its conjugation to glutathione, or it influences phosphorylation dependent signaling pathways leading to increased gene expression, remains to be established. It is worth mentioning that expression of a GST isoform, AtGSTF2 is increased in the *Arabidopsis* mutant *pp2a-b’γ* (see subchapter 3.3.) [29].

#### 2.2.3. Okadaic Acid

OA is one of the most widely used tools for the study of protein phosphatase regulation and functioning. It is a polyether carboxylic acid produced by several Dinoflagellates [41,42]. It inhibits PP2A activity more potently, than PP1 both in vitro and in vivo, but at higher concentrations it has strong inhibitory effects on PP1 activity as well in vivo, including plants [42,79]. OA influences the expression of PP2Ac as well. Interestingly, it increases the level of PP2Ac protein in HeLa and NIH3T3 cells [80].

In spite of its wide usage, the involvement of OA in protein phosphorylation dependent ROS signaling pathways is not well-known for plants. However, there are several studies showing that OA influences markedly MAPKs involved in diverse signaling pathways. For example, OA maintains phosphorylation of the p42 (ERK) and p38 MAPKs in mammalian fibroblasts. This effect is cell type specific [65]. *OsBWMK1*, a JA-responsive MAPK gene of rice can be induced by diverse stresses. A relatively low concentration of OA (1 µM) and high concentrations of cantharidin and endothall resemble the effects of SA, JA, ethylene and ABA as well as H_2_O_2_ in that they increase the transcription level of this gene more rapidly, than ozone. This indicates that regulation mechanisms of MAPK phosphorylation can be both upstream and downstream to ROS production in this case, but an independent mechanism of the inhibition of protein dephosphorylation and ROS production is also possible [81].

As for MCY-LR, OA prevents dephosphorylation of CaMKII. Both inhibitors lead to the activation of this calmodulin-dependent kinase that triggers signaling pathways, resulting in apoptosis of mammalian cells [66].

#### 2.2.4. Lessons from the Effects of PP2A Inhibitors

Inhibitors of PP2A (and PP1) are extensively used for the study of protein dephosphorylation and its role in signaling for plants. Even though there is relatively little knowledge on their PP2A related effects on ROS production and signaling, there are several certain conclusions that can be drawn from the relevant research work. These are summarized on Figure 1. At least for MCY-LR, it is clear that it can generate ROS without the involvement of protein phosphatases by directly conjugating to glutathione through a GST catalyzed mechanism, which induces intracellular redox imbalance. One of the consequences of this is lipid peroxidation. However, all inhibitors for which such information is available inhibit PP2As that dephosphorylate actors of the MAPK cascades. Such signaling cascades are known to be involved in oxidative stress signaling. There is research work that makes probable the acting of these pathways upstream of ROS production. This involves triggering of cascades that lead to the increase of expression and activation of plasma membrane RBOH complex, one of the main ROS generators (Pathways 1 and 3 of oxidative stress signaling, see Introduction and the sections below). To conclude, PP2A inhibitors can be used as excellent tools to reveal this connection between protein phosphorylation and ROS signaling. Their effects will be analyzed in the next sections of the review as well, where they provide relevant information.

## 3. The Involvement of PP2A in Oxidative Stress Pathways

### 3.1. Pathway 1

This pathway involves the signaling events induced by external ROS (see the Introduction section). A recent model shows that apoplastic H_2_O_2_ may activate receptor-like kinases (RLKs) via oxidation of thiol groups. These kinases may actually act as ROS sensors and will then trigger a protein phosphorylation dependent pathway to activate oxidative stress induced transcription factors [82]. The nature of this phosphorylation dependent pathway as well as the protein phosphatases involved are largely unknown. However, a model has been proposed where pathogens or ABA induce stomatal closure, among other mechanisms, by the RBOH-produced ROS, which will activate CRKs (cysteine rich receptor kinases) [83]. It should be noted however, that ABA can also mediate the production of ROS via RBOH through Pathway 3, possibly involving PP2A (See Section 3.3.). Wound or pathogens as well as external H_2_O_2_ induce the expression of OXI1, a serine-threonine kinase, that is further activating MPK3 and MPK6 in *Arabidopsis* [84]. Exogenously applied ozone, H_2_O_2_, and the superoxide radical generating xanthine/xanthine oxidase mixture induced the activation of an ERK-type, Ca^2+^ dependent 46 kDa MAPK in tobacco suspension cells. This activation was obtained in the presence of CA as well, indicating that a PP2A (or PP1) is involved in the (in)activation process [85]. The involvement of pathway 1 in PP2A regulated stress response was suggested by Baier et al. [86] as well: ozone induced extracellular H_2_O_2_ formation was possibly triggering a PP2A regulated intracellular event. Several studies indicate the existence of plasma membrane, cytosolic or endomembrane proteins other than RLKs/CRKs (including the ethylene receptor ETR1) that are “sensors” of external ROS, e.g., oxidation at the level of cysteine thiol groups induces changes in their conformation and as a consequence, their activity. These changes may regulate different signaling pathways [14]. It remains to be established whether these pathways involve PP2A. For wheat, a PP2A catalytic subunit belonging to family II (TaPP2Ac) negatively regulates the expression of genes for the antioxidant enzymes CAT and APX2 as well of the pathogenesis response protein PR2. Infection with the fungal pathogen *Rhizoctonia cerealis* increases the expression of this PP2Ac and exogenous H_2_O_2_ application resembles this effect [39]. As we will see for pathway 3 [29], changes of PP2Ac activities that are due to misfunctioning of B regulatory subunits have partially different effects in ROS signaling, which underlines that the relevant involvement of this phosphatase depends largely on the type of activity change and subcellular localization of these effects. Many actual or possible signaling pathways induced by external H_2_O_2_ (Pathway 1) may be in close correlation with Pathway 3, because apoplastic ROS are produced largely by the plasma membrane RBOH complex that is activated via biotic or abiotic stress induced intracellular responses (see Section 3.3.).

### 3.2. Pathway 2

Pathway 2 involves the signaling events induced by intracellularly generated excess ROS. This is generated e.g., in chloroplasts under excess light conditions. However, the pathways leading to generation of these ROS may be partially linked to Pathways 1 and 3: it has been speculated that ROS generated via RBOH as the endpoint of intracellular signaling, and/or ROS arising from other cells of tissue during systemic oxidative stress- are triggering the production of excess ^1^O_2_, O_2_^•−^ and H_2_O_2_ in chloroplasts [82,87]. During high light intensity conditions OXI1, a cytoplasmic protein kinase is strongly induced by singlet oxygen, the main ROS generated in chloroplasts [88]. Since OXI1 is also induced by external peroxide (see Section 3.1.), it seems that its activity depends on multiple oxidative stress signaling pathways, but it also may have a role in integrating these diverse pathways. It is well-known that the redox sensitive PP2Cs ABI1 and ABI2 are inactivated by H_2_O_2_ (including intracellularily generated peroxide) during ABA signaling and this maintains the phosphorylated-therefore activated- state of MAPKs [89,90,91].

### 3.3. Pathway 3

Regarding this pathway, the connection of ABA signaling- ROS and the protein phosphatase PP2C is well-known. Binding of ABA to its receptor suppresses PP2C activity, leaving the kinase OPEN STOMATA1 (OST1) in a phosphorylated state. This kinase then activates RBOH (plasma membrane NADPH oxidase) at two serine residues to produce ROS in the apoplastic space. This ROS will re-enter the cell, where it triggers mobilization of Ca^2+^ from endomembranes, a major mechanism of ABA induced stomatal closure [92,93,94]. ABA signaling via PP2C activity inhibition, thus activation of protein kinases of the SnRK family (e.g., OST1) will lead to activation of ABA-related gene expression (see reference [93] for a review). An elegant study of reference [40] showed that PP2A may be involved in the same ABA signaling pathway: the *Arabidopsis* PP2A catalytic subunit loss of function mutant *pp2ac-2* showing decreased PP2A activity can partially suppress the ABA-insensitive phenotype of the *abi1-1* gain of function mutant- which shows increased PP2C activity. Interestingly, the *pp2ac-2* mutant showed increased transcription of *pp2ac*, in contrast to inhibition of the enzyme activity itself. Other studies showed that CA, a PP1 and PP2A inhibitor activates RBOH mediated ROS production (reference [91] and references therein).

ROS produced by RBOH in the apoplast increase cytoplasmic Ca^2+^ concentration by opening intracellular Ca^2+^ channels. This increased intracellular [Ca^2+^] will further activate RBOH via binding to it, or by activating Ca^2+^-dependent protein kinases (CDPKs) that phosphorylate RBOH. CPK5, a CDPK, can phosphorylate RBOHD at four serine residues [91,92]. A potato CDPK phosphorylates RBOHB at two serine residues [16]. Thus, RBOHs are susceptible for dephosphorylation by serine-threonine protein phosphatases.

Regarding Pathway 3, further lessons are coming from other PP2A related mutants. *Arabidopsis pp2a-b’γ* is a knockdown mutant of a PP2A B’ subunit. This knockdown inactivates PP2A at the proper subcellular sites of pathogenesis response (PR) [33]. As a consequence, the phenotype will be of a lesion-mimic mutant: even in the absence of pathogen derived signals, leaf chloroplasts tend to be degraded, H_2_O_2_ is produced, while chloroplast Cu-ZnSOD levels increase and GPx, peroxiredoxin Q decrease. Thus, the role of regulatory subunit is the activation of PP2A during normal conditions, that prevents PR. PR, usually induced by SA or JA, involves inactivation of PP2A, that increases the phosphorylation state of Constitutive expression of PR genes 5 (CPR5), leading to H_2_O_2_ production and induction of PR related gene expression via DNA demethylation [33]. This B’ regulatory subunit of PP2A turned to be cytoplasmic, but it has been reported to have an important role in the unfolded protein response at the ER level. The *pp2a-b’γ* mutants were characterized by high phosphorylation level of Calreticulin1 (CRT1), a protein important in the prevention of ER stress. Hyperphosphorylation of this protein inhibited the degradation of unfolded proteins in the ER, leading to cell death [29]. Later on, it turned out that B’γ probably acts at the level of plasmalemma and nucleus as well and the idea of preventing unnecessary PR by PP2A B’γ was further supported [34]. These authors have showed that PP2A prevents SA dependent signaling. SA signaling leads directly or indirectly to the phosphorylation of the PR2, 5 (and probably PR1) proteins as well as of GSTF2: the double mutant *cat2 pp2a-b’γ* showed PR even under non-permissive, short-day conditions (the single catalase mutant *cat2* shows PR only under long-day conditions). Inactivation of RBOH by PP2A was also proposed to be highly probable. According to this scenario, PP2A inactivates RBOH under normal conditions, but during PR, the phosphatase is probably inactivated due to oxidative stress signaling, which leads to ROS production. Meanwhile, stress signals will activate (i) SA dependent cellular events [34], (ii) SA-independent ROS eliminating mechanisms like the increase in the level of mitochondrial alternative oxidases AOX1A and AOX1D, where PP2A plays probably an indirect, negative regulatory role. PP2A B’γ interacts directly with ACONITASE3 in the cytoplasm. This is an important observation, because aconitase, an important enzyme of the citric cycle in mitochondria, is known to control negatively AOX1A [36]. (iii) the ACC oxidase ACO2, which plays an important role in ethylene (ET) production [34]. Regarding ET production, RCN1, a PP2A-Aα scaffolding subunit activates PP2Ac that will inhibit ET biosynthesis and regulate auxin transport to promote normal root elongation growth [30]. ET signaling leads to the induction of defense genes following necrotroph infection that leads to ROS production. SA inhibits this signaling [5]. Since both ET and SA signaling pathways are deactivated by protein dephosphorylation, the effects of PP2A on PR triggered by these two hormones seem to be independent. Regarding PR, a MAPK dependent pathway leads to the activation of transcription factors of the WRKY family. Members of this TF family are activators of SA dependent signaling, since they activate transcription of PR genes and of RBOH [95,96,97]. Since PP2A inhibitors are likely to induce ROS production through activation of MAPK cascades (see Chapter 2 of this review), these observations might add further insights into the possible regulatory roles of protein phosphatases.

Interestingly, the *rcn1* mutant *Arabidopsis* plants showed increased sensitivity to H_2_O_2_, since root growth inhibition in the presence of this ROS was more pronounced in mutants, than in wild-type plants [31]. This shows again the importance of PP2A mediated control of oxidative stress responses. The *rcn1* mutants also show ABA insensitivity [20,98], in contrast to the ABA hypersensitive phenotype of *pp2ac-2* mutants shown above. This indicates that different PP2A heterotrimer combinations may be involved in different steps of the ABA mediated signal transduction pathway. Interestingly, PP2A activation by PP2A B’γ can also trigger stress defense responses: it increases the level of peroxisomal serine:glyoxylate aminotransferase (SGAT), a key enzyme in photorespiration [34] and the transcript level of APX2 [35]. For APX2, it remains to be established whether PP2A mediated increase of its transcription is downstream or upstream to intracellular ROS production (that is, whether PP2A is involved in the regulation of Pathway 2 or Pathway 3 here).

The PP2A B’θ subunit is localized to peroxisomes and negatively regulates plant immunity responses, as suggested by results on *Arabidopsis* knockdown mutants infected by *Pseudomonas syringae* [37]. Silencing of the gene for a type I PP2Ac in *Nicotiana benthamiana* decreased drastically PP2A activity and induced PR (including localized cell death) without any fungal infection. Moreover, these gene silenced plants were more resistant to the bacterial pathogen *Pseudomonas syringae* pv. *tabaci* and the fungal pathogen *Cladosporium fulvum*, showing rapid PR [38]. Since PR involves ROS production, from all information presented in this subsection, we can conclude that PP2A exerts negative regulatory effects on multiple signaling pathways leading to ROS production in plants.

As we can see from the above statements, a large part of the information regarding phosphoregulation of ROS producing proteins refer to RBOH isoforms. However, there are other apoplastic ROS producing proteins like type III peroxidases producing H_2_O_2_ in a pH-dependent manner, germin/oxalate oxidases, amine oxidases etc. [12,99,100]. Little is known about phosphoregulation of these enzymes.

Finally, there is still little knowledge on the PP2A dependent signaling that regulates the production of ROS scavenging enzymes. However, as we have stated before (subchapters 3.1. and this section) there is evidence showing a direct relationship between these phenomena. Wheat plants with a silenced catalytic subunit of PP2A (TaPP2Ac) show increased expression of PR2 as well as CAT and APX isoforms indicating that PP2A negatively regulates plant defense responses [34]. Regulatory subunits of PP2A may have contrary effects in different plant models, as the case of the *Arabidopsis pp2a-b’γ*, a lesion-mimic mutant, where ROS scavenging enzymes are strongly de-regulated [34].

### 3.4. Concluding Remarks

Oxidative stress signaling includes pathways leading to cell death and defense against biotic and abiotic stressors. In general, ROS can be induced by a wide range of intra- and extracellular events. Therefore, it is not surprising that pathways regarding ROS signaling are diverse—see the three pathways presented in Section 1 and Figure 2. Moreover, these pathways are interconnected. For example, RBOH produced and activated by Pathway 3 will produce ROS in the apoplast, which will then trigger Pathway 1, for example by the cell-to cell ROS wave that induces systemic responses e.g., to pathogens.

The involvement of PP2C in the ABA induced ROS signaling is well known: this phosphatase is inactivated at ABA exposure of cell (e.g., the stomatal guard cell), that will increase phosphorylation state of proteins, leading finally to changes in gene expression. This inactivation is triggered by binding of ABA to the PYR/RCAR receptor, which will separate ABI1 (PP2C) from its substrates and triggers Pathway 3. Pathway 2 also triggers PP2C inactivation through its direct interaction with intracellularly produced ROS. Recently, even though there are many uncertainties, there is more and more evidence that PP2A is also an important player in sensing and transducing oxidative stress in plants. These roles are summarized on Figure 2. As we have seen in Section 2, PP2A inhibitors have already pointed out the role of protein kinases/ kinase cascades in these phenomena and made the involvement of PP2A highly possible. Other experimental approaches, like the use of PP2A related mutants, added further proof for the role of this phosphatase in the control of protein kinase dependent stress signaling pathways. The central player of ROS signaling-RBOH- is dependent on PP2A at multiple levels. This phosphatase can exert such an effect through B’γ and B’θ regulatory (in this case, activator) subunits. Through them, the enzyme plays a negative controlling effect on oxidative stress responses under normal, non-stressful conditions, preventing the production of excess ROS. Wounding, pathogens, abiotic stresses trigger cellular responses through ABA, SA and JA mediated pathways, in many cases when PP2A is inactivated and the phosphorylation occurs, the activation state of stress signaling proteins via protein kinases will increase, leading to increased ROS production and the modulation of ROS scavenging enzyme levels-and in consequence to cell death and/or defense responses. What are the mechanisms of this phosphatase inactivation? This is an interesting topic for future research.

## Figures and Tables

**Figure 1 ijms-20-03028-f001:**
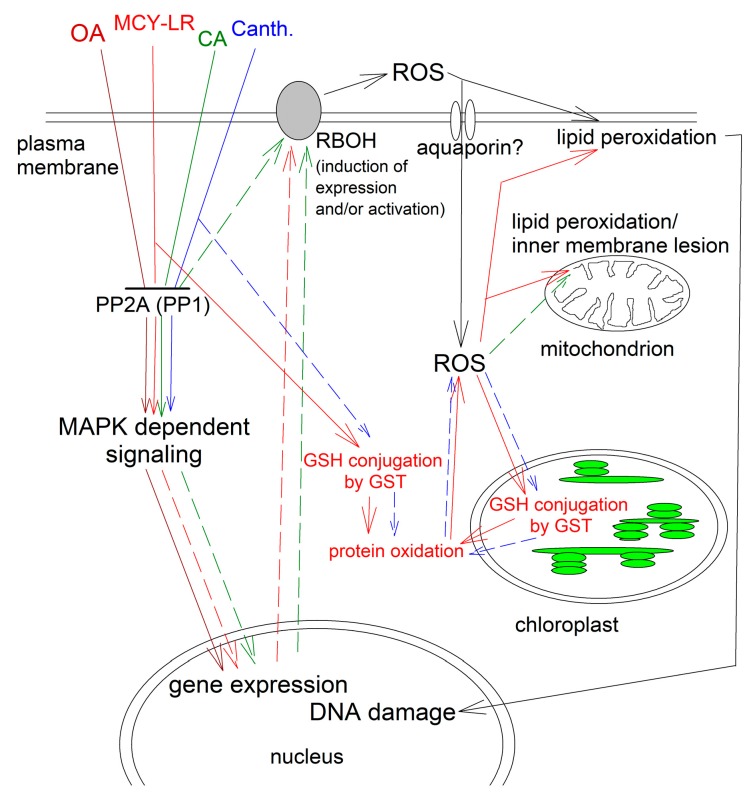
ROS-related targets of PP2A inhibitors in plants. Clear lines/arrows show mechanisms that are already elucidated, while dashed lines/arrows show mechanisms for which some evidence exists, but further research is needed for clarifying. As can be seen, all inhibitors activate MAPK cascades by PP2A inhibition and this might be a key step in the production of ROS via RBOH. On the other hand, a decrease in reduced glutathione (GSH) pool due to the formation of GSH-MCY-LR (or cantharidin) conjugates leads to the elevation of ROS levels in a PP2A independent pathway. For MCY-LR, there are three possible mechanisms of oxidative stress induction (see text) of which mechanisms 1 and 3 that have been proven for plants are presented here. Lipid peroxidation and protein oxidation can originate from both mechanisms as shown in the Figure, thus a clear separation of these mechanisms is difficult. It was not our scope to show here the uptake mechanisms of inhibitors by plasma membrane and endomembranes. GSH: reduced glutathione; MCY-LR: microcystin-LR; CA: calyculin A; OA: okadaic acid. The effects of different inhibitors are shown as red: MCY-LR; green: CA; blue: cantharidin; brown: OA.

**Figure 2 ijms-20-03028-f002:**
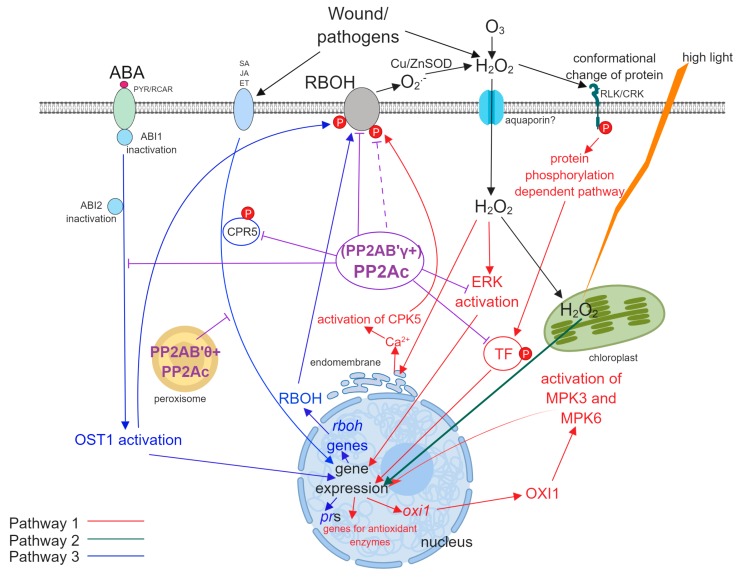
An overview of proven and hypothetical oxidative stress signaling pathways, where PP2A is involved. Red: Pathway 1; green: Pathway 2; Blue: Pathway 3. Clear lines/arrows show pathways that are already elucidated, while dashed lines/arrows show pathways for which some evidence exists, but further research is needed for clarification. The central player of these players is RBOH: the pathway of its expression and activation is controlled at multiple steps of Pathway 3 and it is also crucial for the production of apoplastic ROS, that will then re-enter Pathway 1 that is also controlled by PP2A. There is poor information on the regulation of Pathway 2 by PP2A. CPR5 is a phosphoregulated protein involved in SA signaling. CPK5 and ERK are Ca^2+^ activated protein kinases. OST1 (OPEN STOMATA 1) is a protein kinase involved in the activation of RBOH. OXI1 is a Ser/thr kinase and RLK/CRK is a large family of receptor-like kinases. PYR/RCAR is the ABA receptor and ABI1 and ABI2 are the PP2Cs inactivated by ABA.

**Table 1 ijms-20-03028-t001:** Some important examples of A (scaffolding), B (regulatory) and C (catalytic) subunits of plant PP2A complexes in relation to oxidative stress responses.

Subunit	Gene/Mutant Name	Organism	Function: Activation/Inactivation of PP2A/C (for the A and B Subunits) and Physiological Consequence	Ref.
PP2A/Aα	*rcn1*	*Arabidopsis thaliana*	Activates PP2Ac that will inhibit ET biosynthesis and regulate auxin transport to promote normal root elongation growth. It confers ABA sensitivity. Loss of function of *rcn1* promotes ET signaling and leads to the induction of defence genes following necrotroph infection that leads to ROS production.	[30,31]
PP2A/A3	*pp2a/a3*	*Arabidopsis thaliana*	Interacts with the E3 ubiquitin ligase AtCHIP, which will increase PP2A catalytic activity that modulates responses to dark and cold treatments as well as ABA sensitivity. AtCHIP overexpressing plants are cold sensitive.	[32]
PP2A/B′γ	*pp2a-b’γ*	*Arabidopsis thaliana*	Activation of PP2A. Pathogenesis response(PR), usually induced by SA or JA, involves inactivation of PP2A e.g., by inactivation of this regulatory subunit, that increases the phosphorylation state of CONSTITUTIVE EXPRESSION OF PR GENES5 (CPR5), leading to H_2_O_2_ production and induction of PR related gene expression via DNA demethylation. B’γ is also important in the regulation of peroxisomal serine:glyoxylate aminotransferase (SGAT) activity, expression and RBOH activation, expression of a GST isoform and of APX2. It has probably an indirect role in the regulation of ET biosynthesis.	[29,33,34,35]
PP2A/B′*γ* and PP2A/B′*ζ*	*pp2a-b’γ and pp2a-b’ ζ*	*Arabidopsis thaliana*	Activate PP2A, keep stress tolerance enzymes at low level under normal (non-stressed) conditions. PP2A-B′*γ* negatively regulates the mitochondrial alternative oxidases AOX1A and AOX1D	[35,36]
PP2A/B’θ	*b’θ-1*	*Arabidopsis thaliana*	Activates PP2A, localized to peroxisomes and negatively regulates plant immunity responses.	[37]
PP2A/C	*PP2Ac*	*Nicotiana benthamiana*	Inhibition of PR under normal (non-stressed) conditions. During pathogen infection, its gene is probably silenced, conferring resistance to bacterial and fungal pathogens.	[38]
PP2A/C, family II	*TaPP2Ac*	wheat	Decreases expression of CAT, APX2 and PR2. Infection with the fungal pathogen *Rhizoctonia cerealis* and H_2_O_2_ increases its expression.	[39]
PP2A/C	*pp2ac-2*	*Arabidopsis thaliana*	The loss of function mutant phenotype can partially suppress the ABA-insensitive phenotype of the *abi1-1* gain of function mutant - which shows increased PP2C activity.	[40]

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
