# Peer review of "The Role of Serine-Threonine Protein Phosphatase PP2A in Plant Oxidative Stress Signaling—Facts and Hypotheses"

_ijms, 2019, doi:10.3390/ijms20123028_

Reviewer 1 Report

This review is an overview of possible roles of serine-threonine protein phosphatases in plant oxidative stress signaling with particular attention on PP2A. The topic is very interesting since these mechanisms have implication in the understanding of stress tolerance of vascular plants.

The review is well written in a fluent language and the figures help to follow the paper.

I suggest accepting the manuscript in the present form.

Author Response

We would like to thank Reviewer 1 for considering our manuscript. We hope the paper will be useful for those interested in plant oxidative stress.

Reviewer 2 Report

I would like to suggest some modifications for the publication in this journal. Please address comments below.

In section 2.1, it will be good to summarize information of PP2As on the table or figs. In this section, characteristics of different PP2A are introduced and the information described in this part should be important to understand the later part of this review.

In the section 2.2.1, Model 1-3 are explained in detail and Figure 1 was mentioned. Authors should therefore indicate which pathway are related to Model 1-3 in the Figure 1.

On lines 299-303, functions of PHB and CDR proteins should be explained more. It should be helpful to understand the pathways addressed in this part.

 On line 353 and lines 465-466, more detailed explanations are required. In these parts, authors just mention involvement of CaMKII and UPR at the ER in the phosphorylation-dependent processes. Authors should explain more detailed pathways.

Please also check the minor point below

On line 42 and 226, it should be better not to use the term “oxidative damage” or “oxidative stress”. On line 42, oxidative burst via RBOH proteins is positive mechanism that can protect plants against biotic stresses. In this case, the important point is to mention that “dramatic increase of ROS production” ("not oxidative damage") can be induced. On line 226, I understood what authors want to appeal. But, in general, ROS production itself is not equal to oxidative stress. In this case, authors should say “excess” ROS production can lead to lipid peroxidation and other damaging processes. In some cases, “ROS production” and “oxidative stress” should be distinguished to prevent misleading for readers who are not expert of ROS regulatory systems.

Author Response

Reviewer 2

 We would like to thank Reviewer 2 for his/her useful comments that helped in improving the manuscript. Please find below our detailed answers to the comments.

„In section 2.1, it will be good to summarize information of PP2As on the table or figs. In this section, characteristics of different PP2A are introduced and the information described in this part should be important to understand the later part of this review.”

We prepared a table to summarize the role of PP2A subunits in oxidative stress response/signaling.

In the section 2.2.1, Model 1-3 are explained in detail and Figure 1 was mentioned. Authors should therefore indicate which pathway are related to Model 1-3 in the Figure 1.”

For more accuracy, we have now stated in the caption of Figure 1: “For MCY-LR, there are three possible mechanisms of oxidative stress induction (see text) of which mechanisms 1 and 3, proven for plants are presented here. Lipid peroxidation and protein oxidation can originate from both mechanisms as shown in the Figure, thus a clear separation of these mechanisms is difficult.”

"On lines 299-303, functions of PHB and CDR proteins should be explained more. It should be helpful to understand the pathways addressed in this part."

New information was added to the manuscript in order to make the functions of these proteins more clear (see lines 291-296 of revision).

"On line 353 and lines 465-466, more detailed explanations are required. In these parts, authors just mention involvement of CaMKII and UPR at the ER in the phosphorylation-dependent processes. Authors should explain more detailed pathways."

All this information has been added in the revision, please check lines 347-349 and 472-475 of revision.

“Please also check the minor point below

On line 42 and 226, it should be better not to use the term “oxidative damage” or “oxidative stress”. On line 42, oxidative burst via RBOH proteins is positive mechanism that can protect plants against biotic stresses. In this case, the important point is to mention that “dramatic increase of ROS production” ("not oxidative damage") can be induced. On line 226, I understood what authors want to appeal. But, in general, ROS production itself is not equal to oxidative stress. In this case, authors should say “excess” ROS production can lead to lipid peroxidation and other damaging processes. In some cases, “ROS production” and “oxidative stress” should be distinguished to prevent misleading for readers who are not expert of ROS regulatory systems.”

We followed a broader definition of oxidative stress:

“Oxidative stress results from conditions that promote formation of ROS, which can damage or kill cells. Environmental factors that cause oxidative stress (Fig. 22.33) include air pollution (increased amounts of ozone or sulfur dioxide), oxidant-forming herbicides such as Paraquat (methyl viologen, 1,1’-dimethyl-4,4’ bipyridinium), heavy metals, drought, heat and cold stress, wounding, the transition to anoxia and reoxygenation, UV light, and intense light conditions that stimulate photoinhibition (see Chapter 12). Oxidative stress also occurs in response to senescence (see Chapter 20) and pathogen infection (see Chapter 21).”

Paragraph was cited from: Shinozaki, K.; Uemura, M.; Bailey-Serres, J.; Bray, E.A.; Weretilnyk, E. Responses to Abiotic Stress. In Biochemistry and Molecular Biology of Plants; Buchanan, B.B., Gruissem, W., Jones, R.L., Eds.; John Wiley & Sons, 2015; pp. 1085–1094 ISBN 978-0-470-71422-5.

Reviewer 3 Report

This review is well written and discuss a very unique issue. However, the auhorsare confined only three pathways. I suggest the authors discuss the rlatiion with oxidative stress, ROS metabolism and signaling.

Also, summerize how these protens helps plant in surviving ubder abiotic stress.

Author Response

Reviewer 3

We would like to thank Reviewer 3 for his/her useful comments to the manuscript. Remarks were taken into consideration when we prepared the revised version.

„Comments and Suggestions for Authors

This review is well written and discuss a very unique issue. However, the auhorsare confined only three pathways. I suggest the authors discuss the rlatiion with oxidative stress, ROS metabolism and signaling.

Also, summerize how these protens helps plant in surviving ubder abiotic stress.”

As we now state repeatedly in the revision, besides the three pathways of ROS signaling presented, the scavenging of ROS is also very important. Although the involvement of the direct relationship between PP2A and plant ROS defense is not well-known, we summarize some evidence for this at the end of subchapter 3.3.

Regarding survival of plants during different stress types, the final thoughts of the Conclusion section are: “Wounding, pathogens, abiotic stresses trigger cellular responses through ABA, SA and JA mediated pathways, when PP2A is inactivated and the phosphorylation- in many cases, the activation state- of stress signaling proteins via protein kinases, will increase, leading to increased ROS production and the modulation of ROS scavenging enzyme levels-and in consequence to cell death and/or defense responses. What are the mechanisms of this phosphatase inactivation? This is an interesting topic of future research.”

Round  2

Reviewer 2 Report

The current version of the manuscript is sufficient for the publication. Authors properly answered the comments and improved the manuscript.

Reviewer 3 Report

The revised version if fine to me.